# Auditing: Active Learning with Outcome-Dependent Query Costs

**Sivan Sabato**
Microsoft Research New England
sivan.sabato@microsoft.com

**Anand D. Sarwate**
TTI-Chicago
asarwate@ttic.edu

**Nathan Srebro**
Technion-Israel Institute of Technology and TTI-Chicago
nati@ttic.edu

## Abstract

We propose a learning setting in which unlabeled data is free, and the cost of a label depends on its value, which is not known in advance. We study binary classification in an extreme case, where the algorithm only pays for negative labels. Our motivation are applications such as fraud detection, in which investigating an honest transaction should be avoided if possible. We term the setting *auditing*, and consider the *auditing complexity* of an algorithm: the number of negative labels the algorithm requires in order to learn a hypothesis with low relative error. We design auditing algorithms for simple hypothesis classes (thresholds and rectangles), and show that with these algorithms, the auditing complexity can be significantly lower than the active label complexity. We also show a general competitive approach for learning with outcome-dependent costs.

## 1 Introduction

Active learning algorithms seek to mitigate the cost of learning by using unlabeled data and sequentially selecting examples to query for their label to minimize total number of queries. In some cases, however, the actual cost of each query depends on the true label of the example and is thus *not known* before the label is requested. For instance, in detecting fraudulent credit transactions, a query with a positive answer is not wasteful, whereas a negative answer is the result of a wasteful investigation of an honest transaction, and perhaps a loss of good-will. More generally, in a multiclass setting, different queries may entail different costs, depending on the outcome of the query. In this work we focus on the binary case, and on the extreme version of the problem, as described in the example of credit fraud, in which the algorithm only pays for queries which return a negative label. We term this setting *auditing*, and the cost incurred by the algorithm its *auditing complexity*.

There are several natural ways to measure performance for auditing. For example, we may wish the algorithm to maximize the number of positive labels it finds for a fixed "budget" of negative labels, or to minimize the number of negative labels while finding a certain number or fraction of positive labels. In this work we focus on the classical learning problem, in which one attempts to learn a classifier from a fixed hypothesis class, with an error close to the best possible. Similar to active learning, we assume we are given a large set of unlabeled examples, and aim to learn with minimal labeling cost. But unlike active learning, we only incur a cost when requesting the label of an example that turns out to be negative.

The close relationship between auditing and active learning raises natural questions. Can the auditing complexity be significantly better than the label complexity in active learning? If so, should

algorithms be optimized for auditing, or do optimal active learning algorithms also have low auditing complexity? To answer these questions, and demonstrate the differences between active learning and auditing, we study the simple hypothesis classes of thresholds and of axis-aligned rectangles in $\mathbb{R}^d$, in both the realizable and the agnostic settings. We then also consider a general competitive analysis for arbitrary hypothesis classes.

**Other work.** Existing work on active learning with costs (Margineantu, 2007; Kapoor et al., 2007; Settles et al., 2008; Golovin and Krause, 2011) typically assumes that the cost of labeling each point is known *a priori*, so the algorithm can use the costs directly to select a query. Our model is significantly different, as the costs depend on the outcome of the query itself. Kapoor et al. (2007) do mention the possibility of class-dependent costs, but this possibility is not studied in detail. An unrelated game-theoretic learning model addressing "auditing" was proposed by Blocki et al. (2011).

### Notation and Setup

For an integer $m$, let $[m] = \{1, 2, \ldots, m\}$. The function $\mathbb{I}[A]$ is the indicator function of a set $A$. For a function $f$ and a sub-domain $X$, $f|_X$ is the restriction of $f$ to $X$. For vectors $\mathbf{a}$ and $\mathbf{b}$ in $\mathbb{R}^d$, the inequality $\mathbf{a} \leq \mathbf{b}$ implies $a_i \leq b_i$ for all $i \in [d]$.

We assume a data domain $\mathcal{X}$ and a distribution $D$ over labeled data points in $\mathcal{X} \times \{-1, +1\}$. A learning algorithm may sample i.i.d. pairs $(X, Y) \sim D$. It then has access to the value of $X$, but the label $Y$ remains hidden until queried. The algorithm returns a labeling function $\hat{h} : \mathcal{X} \to \{-1, +1\}$. The error of a function $h : \mathcal{X} \to \{-1, +1\}$ on $D$ is $\mathrm{err}(D, h) = \mathbb{E}_{(X,Y) \sim D}[h(X) \neq Y]$. The error of $h$ on a multiset $S \subseteq \mathcal{X} \times \{-1, +1\}$ is given by $\mathrm{err}(S, h) = \frac{1}{|S|} \sum_{(x,y) \in S} \mathbb{I}[h(x) \neq y]$. The *passive sample complexity* of an algorithm is the number of pairs it draws from $D$. The *active label complexity* of an algorithm is the total number of label queries the algorithm makes. Its *auditing complexity* is the number of queries the algorithm makes on points with negative labels.

We consider guarantees for learning algorithms relative to a hypothesis class $\mathcal{H} \subseteq \{-1, +1\}^{\mathcal{X}}$. We denote the error of the best hypothesis in $\mathcal{H}$ on $D$ by $\mathrm{err}(D, \mathcal{H}) = \min_{h \in \mathcal{H}} \mathrm{err}(D, h)$. Similarly, $\mathrm{err}(S, \mathcal{H}) = \min_{h \in \mathcal{H}} \mathrm{err}(S, h)$. We usually denote the best error for $D$ by $\eta = \mathrm{err}(D, \mathcal{H})$.

To describe our algorithms it will be convenient to define the following sample sizes, using universal constants $C, c > 0$. Let $\delta \in (0, 1)$ be a confidence parameter, and let $\epsilon \in (0, 1)$ be an error parameter. Let $m^{\mathrm{ag}}(\epsilon, \delta, d) = C(d + \ln(c/\delta))/\epsilon^2$. If a sample $S$ is drawn from $D$ with $|S| = m^{\mathrm{ag}}(\epsilon, \delta, d)$ then with probability $1 - \delta$, $\forall h \in \mathcal{H}$, $\mathrm{err}(D, h) \leq \mathrm{err}(S, h) + \epsilon$ and $\mathrm{err}(S, \mathcal{H}) \leq \mathrm{err}(D, \mathcal{H}) + \epsilon$ (Bartlett and Mendelson, 2002). Let $m_\nu(\epsilon, \delta, d) = C(d \ln(c/\nu\epsilon) + \ln(c/\delta))/\nu^2\epsilon$. Results of Vapnik and Chervonenkis (1971) show that if $\mathcal{H}$ has VC dimension $d$ and $S$ is drawn from $D$ with $|S| = m_\nu$, then for all $h \in \mathcal{H}$,

$$\mathrm{err}(S, h) \leq \max\{\mathrm{err}(D, h)(1 + \nu), \mathrm{err}(D, h) + \nu\epsilon\} \text{ and} \tag{1}$$
$$\mathrm{err}(D, h) \leq \max\{\mathrm{err}(S, h)(1 + \nu), \mathrm{err}(S, h) + \nu\epsilon\}.$$

## 2   Active Learning vs. Auditing: Summary of Results

The main point of this paper is that the auditing complexity can be quite different from the active label complexity, and that algorithms tuned to minimizing the audit label complexity give improvements over standard active learning algorithms. Before presenting these differences, we note that in some regimes, neither active learning nor auditing can improve significantly over the passive sample complexity. In particular, a simple adaptation of a result of Beygelzimer et al. (2009), establishes the following lower bound.

**Lemma 2.1.** *Let $\mathcal{H}$ be a hypothesis class with VC dimension $d > 1$. If an algorithm always finds a hypothesis $\hat{h}$ with $\mathrm{err}(D, \hat{h}) \leq \mathrm{err}(D, \mathcal{H}) + \epsilon$ for $\epsilon > 0$, then for any $\eta \in (0, 1)$ there is a distribution $D$ with $\eta = \mathrm{err}(D, \mathcal{H})$ such that the auditing complexity of this algorithm for $D$ is $\Omega(d\eta^2/\epsilon^2)$.*

That is, when $\eta$ is fixed while $\epsilon \to 0$, the auditing complexity scales as $\Omega(d/\epsilon^2)$, similar to the passive sample complexity. Therefore the two situations which are interesting are the realizable

case, corresponding to $\eta = 0$, and the agnostic case, when we want to guarantee an excess error $\epsilon$ such that $\eta/\epsilon$ is bounded. We provide results for both of these regimes.

We will first consider the *realizable case*, when $\eta = 0$. Here it is sufficient to consider the case where a fixed pool $S$ of $m$ points is given and the algorithm must return a hypothesis $\hat{h}$ such that $\text{err}(S, \hat{h}) = 0$ with probability 1. A pool labeling algorithm can be used to learn a hypothesis which is good for a distribution by drawing and labeling a large enough pool. We define auditing complexity for an unlabeled pool as the minimal number of negative labels needed to perfectly classify it. It is easy to see that there are pools with an auditing complexity at least the VC dimension of the hypothesis class.

For the *agnostic case*, when $\eta > 0$, we denote $\alpha = \epsilon/\eta$ and say that an algorithm $(\alpha, \delta)$-learns a class of distributions $\mathcal{D}$ with respect to $\mathcal{H}$ if for all $D \in \mathcal{D}$, with probability $1 - \delta$, $\hat{h}$ returned by the algorithm satisfies $\text{err}(D, \hat{h}) \leq (1 + \alpha)\eta$. By Lemma 2.1 an auditing complexity of $\Omega(d/\alpha^2)$ is unavoidable, but we can hope to improve over the passive sample complexity lower bound of $\Omega(d/\eta\alpha^2)$ (Devroye and Lugosi, 1995) by avoiding the dependence on $\eta$.

Our main results are summarized in Table 1, which shows the auditing and active learning complexities in the two regimes, for thresholds on $[0, 1]$ and axis-aligned rectangles in $\mathbb{R}^d$, where we assume that the hypotheses label the points in the rectangle as negative and points outside as positive.

|  |  | **Active** | **Auditing** |
| --- | --- | --- | --- |
| **Realizable** | Thresholds | $\Theta(\ln m)$ | 1 |
|  | Rectangles | $m$ | $2d$ |
| **Agnostic** | Thresholds | $\Omega\left(\ln\left(\frac{1}{\eta}\right) + \frac{1}{\alpha^2}\right)$ | $O\left(\frac{1}{\alpha^2}\right)$ |
|  | Rectangles | $\Omega\left(d\left(\frac{1}{\eta} + \frac{1}{\alpha^2}\right)\right)$ | $O\left(d^2 \ln^2\left(\frac{1}{\eta}\right) \cdot \frac{1}{\alpha^2} \ln\left(\frac{1}{\alpha}\right)\right)$ |

Table 1: Auditing complexity upper bounds vs. active label complexity lower bounds for realizable (pool size $m$) and agnostic ($\text{err}(D, \mathcal{H}) = \eta$) cases. Agnostic bounds are for $(\alpha, \delta)$-learning with a fixed $\delta$, where $\alpha = \epsilon/\eta$.

In the realizable case, for thresholds, the optimal active learning algorithm performs binary search, resulting in $\Omega(\ln m)$ labels in the worst case. This is a significant improvement over the passive label complexity of $m$. However, a simple auditing procedure that scans from right to left queries only a single negative point, achieving an auditing complexity of 1. For rectangles, we present a simple coordinate-wise scanning procedure with auditing complexity of at most $2d$, demonstrating a huge gap versus active learning, where the labels of all $m$ points might be required. Not all classes enjoy reduced auditing complexity: we also show that for rectangles with positive points on the *inside*, there exists pools of size $m$ with an auditing complexity of $m$.

In the agnostic case we wish to $(\alpha, \delta)$-learn distributions with a true error of $\eta = \text{err}(D, \mathcal{H})$, for constant $\alpha, \delta$. For active learning, it has been shown that in some cases, the $\Omega(d/\eta)$ passive sample complexity can be replaced by an exponentially smaller $O(d\ln(1/\eta))$ active label complexity (Hanneke, 2011), albeit sometimes with a larger polynomial dependence on $d$. In other cases, an $\Omega(1/\eta)$ dependence exists also for active learning. Our main question is whether the dependence on $\eta$ in the active label complexity can be further reduced for auditing.

For thresholds, active learning requires $\Omega(\ln(1/\eta))$ labels (Kulkarni et al., 1993). Using auditing, we show that the dependence on $\eta$ can be completely removed, for any true error level $\eta > 0$, *if we know $\eta$ in advance*. We also show that if $\eta$ is not known at least approximately, the logarithmic dependence on $1/\eta$ is unavoidable also for auditing. For rectangles, we show that the active label complexity is at least $\Omega(d/\eta)$. In contrast, we propose an algorithm with an auditing complexity of $O(d^2 \ln^2(1/\eta))$, reducing the linear dependence on $1/\eta$ to a logarithmic dependence. We do not know whether a linear dependence on $d$ is possible with a logarithmic dependence on $1/\eta$.

Omitted proofs of results below are provided in the extended version of this paper (Sabato et al., 2013).

# 3 Auditing for Thresholds on the Line

The first question to ask is whether the audit label complexity can ever be significantly smaller than the active or passive label complexities, and whether a different algorithm is required to achieve this improvement. The following simple case answers both questions in the affirmative. Consider the hypothesis class of *thresholds on the line*, defined over the domain $\mathcal{X} = [0,1]$. A hypothesis with threshold $a$ is $h_a(x) = \mathbb{I}[x - a \geq 0]$. The hypothesis class is $\mathcal{H}_{\dashv} = \{h_a \mid a \in [0,1]\}$. Consider the pool setting for the realizable case. The optimal active label complexity of $\Theta(\log_2 m)$ can be achieved by a binary search on the pool. The auditing complexity of this algorithm can also be as large as $\Theta(\log_2(m))$. However, auditing allows us to beat this barrier. This case exemplifies an interesting contrast between auditing and active learning. Due to information-theoretic considerations, any algorithm which learns an unlabeled pool $S$ has an active label complexity of at least $\log_2 |\mathcal{H}|_S|$ (Kulkarni et al., 1993), where $\mathcal{H}|_S$ is the set of restrictions of functions in $\mathcal{H}$ to the domain $S$. For $\mathcal{H}_{\dashv}$, $\log_2 |\mathcal{H}_{\dashv}|_S| = \Omega(\log_2 m)$. However, the same considerations are invalid for auditing.

We showed that for the realizable case, the auditing label complexity for $\mathcal{H}_{\dashv}$ is a constant. We now provide a more complex algorithm that guarantees this for $(\alpha, \delta)$-learning in the agnostic case. The intuition behind our approach is that to get the optimal threshold in a pool with at most $k$ errors, we can query from highest to lowest until observing $k + 1$ negative points and then find the minimal error threshold on the labeled points.

**Lemma 3.1.** *Let $S$ be a pool of size $m$ in $[0,1]$, and assume that $\mathrm{err}(S, \mathcal{H}_{\dashv}) \leq k/m$. Then the procedure above finds $\hat{h}$ such that $\mathrm{err}(S, \hat{h}) = \mathrm{err}(S, \mathcal{H}_{\dashv})$ with an auditing complexity of $k + 1$.*

*Proof.* Denote the last queried point by $x_0$, and let $h_{a^*} = \operatorname{argmin}_{h \in \mathcal{H}_{\dashv}} \mathrm{err}(S, \mathcal{H}_{\dashv})$. Since $\mathrm{err}(S, h_{a^*}) \leq k/m$, $a^* > x_0$. Denote by $S' \subseteq S$ the set of points queried by the procedure. For any $a > x_0$, $\mathrm{err}(S', h_a) = \mathrm{err}(S, h_a) + |\{(x, y) \in S \mid x < x_0, y = 1\}|/m$. Therefore, minimizing the error on $S'$ results in a hypothesis that minimizes the error on $S$. $\qquad\square$

To learn from a distribution, one can draw a random sample and use it as the pool in the procedure above. However, the sample size required for passive $(\alpha, \delta)$-learning of thresholds is $\Omega(\ln(1/\eta)/\eta)$. Thus, the number of errors in the pool would be $k = \eta \cdot \Omega(\ln(1/\eta)/\eta) = \Omega(\ln(1/\eta))$, which depends on $\eta$. To avoid this dependence, the auditing algorithm we propose uses Alg. 1 below to select a subset of the random sample, which still represents the distribution well, but its size is only $\Omega(1/\eta)$.

**Lemma 3.2.** *Let $\delta, \eta_{\max} \in (0, 1)$. Let $S$ be a pool such that $\mathrm{err}(S, \mathcal{H}_{\dashv}) \leq \eta_{\max}$. Let $S_q$ be the output of Alg. 1 with inputs $S, \eta_{\max}, \delta$, and let $\hat{h} = \operatorname{argmin}_{h \in \mathcal{H}_{\dashv}} \mathrm{err}(S_q, \mathcal{H}_{\dashv})$. Then with probability $1 - \delta$,*

$$\mathrm{err}(S_q, \hat{h}) \leq 6\eta_{\max} \quad and \quad \mathrm{err}(S, \hat{h}) \leq 17\eta_{\max}.$$

The algorithm for auditing thresholds on the line in the agnostic case is listed in Alg. 2. This algorithm first achieves $(C, \delta)$ learning of $\mathcal{H}_{\dashv}$ for a fixed $C$ (in step 7, based on Lemma 3.2 and Lemma 3.1), and then improves its accuracy to achieve $(\alpha, \delta)$-learning for $\alpha > 0$, by additional passive sampling in a restricted region. The following theorem provides the guarantees for Alg. 2.

---

**Algorithm 1:** Representative Subset Selection

1: **Input:** pool $S = (x_1, \ldots, x_m)$ (with hidden labels), $x_i \in [0,1]$, $\eta_{\max} \in (0,1]$, $\delta \in (0,1)$.
2: $T \leftarrow \max\{\lfloor 1/3\eta_{\max} \rfloor, 1\}$.
3: Let $U = \{\underbrace{x_1, \ldots, x_1}_{T \text{ copies}}, \ldots, \underbrace{x_m, \ldots, x_m}_{T \text{ copies}}\}$ be the multiset with $T$ copies of each point in $S$.
4: Sort and rename the points in $U$ such that $x'_i \leq x'_{i+1}$ for all $i \in [Tm]$.
5: Let $S_q$ be an empty multiset.
6: **for** $t = 1$ to $T$ **do**
7: $\quad S(t) \leftarrow \{x'_{(t-1)m+1}, \ldots, x'_{tm}\}$.
8: $\quad$ Draw $14 \ln(8/\delta)$ random points from $S(t)$ independently uniformly at random and add them to $S_q$ (with duplications).
9: **end for**
10: Return $S_q$ (with the corresponding hidden labels).

---

---
**Algorithm 2:** Auditing for Thresholds with a constant $\alpha$
---
1: **Input:** $\eta_{\max}, \delta, \alpha \in (0, 1)$, access to distribution $D$ such that $\mathrm{err}(D, \mathcal{H}_\dashv) \leq \eta_{\max}$.
2: $\nu \leftarrow \alpha/5$.
3: Draw a random labeled pool (with hidden labels) $S_0$ of size $m_\nu(\eta, \delta/2, 1)$ from $D$.
4: Draw a random sample $S$ of size $m^{\mathrm{ag}}((1 + \nu)\eta_{\max}, \delta/2, 1)$ uniformly from $S_0$.
5: Get a subset $S_q$ using Alg. 1 with inputs $S, 2(1 + \nu)\eta_{\max}, \delta/2$.
6: Query points in $S_q$ from highest to lowest. Stop after $\lceil 12|S_q|(1 + \nu)\eta_{\max}\rceil + 1$ negatives.
7: Find $\hat{\mathbf{a}}$ such that $h_{\hat{\mathbf{a}}}$ minimizes the error on the labeled part of $S_q$.
8: Let $S_1$ be the set of the $36(1 + \nu)\eta_{\max}|S_0|$ closest points to $\hat{\mathbf{a}}$ in $S$ from each side of $\hat{\mathbf{a}}$.
9: Draw $S_2$ of size $m^{\mathrm{ag}}(\nu/72, \delta/2, 1)$ from $S_1$ (see definition on page 2).
10: Query all points in $S_2$, and return $\hat{h}$ that minimizes the error on $S_2$.
---

**Theorem 3.3.** *Let $\eta_{\max}, \delta, \alpha \in (0, 1)$. Let $D$ be a distribution with error $\mathrm{err}(D, \mathcal{H}_\dashv) \leq \eta_{\max}$. Alg. 2 with input $\eta_{\max}, \delta, \alpha$ has an auditing complexity of $O(\ln(1/\delta)/\alpha^2)$, and returns $\hat{h}$ such that with probability $1 - \delta$, $\mathrm{err}(D, \hat{h}) \leq (1 + \alpha)\eta_{\max}$.*

It immediately follows that if $\eta = \mathrm{err}(D, \mathcal{H})$ is known, $(\alpha, \delta)$-learning is achievable with an auditing complexity that does not depend on $\eta$. This is formulated in the following corollary.

**Corollary 3.4** $((\alpha, \delta)$-learning for $\mathcal{H}_\dashv)$**.** *Let $\eta, \alpha, \delta \in (0, 1]$. For any distribution $D$ with error $\mathrm{err}(D, \mathcal{H}_\dashv) = \eta$, Alg. 2 with inputs $\eta_{\max} = \eta, \alpha, \delta$ $(\alpha, \delta)$-learns $D$ with respect to $\mathcal{H}_\dashv$ with an auditing complexity of $O(\ln(1/\delta)/\alpha^2)$.*

A similar result holds if the error is known up to a multiplicative constant. But what if no bound on $\eta$ is known? The following lower bound shows that in this case, the best active complexity for threshold this similar to the best active label complexity.

**Theorem 3.5** (Lower bound on auditing $\mathcal{H}_\dashv$ without $\eta_{\max}$)**.** *Consider any constant $\alpha \geq 0$. For any $\delta \in (0, 1)$, if an auditing algorithm $(\alpha, \delta)$-learns any distribution $D$ such that $\mathrm{err}(D, \mathcal{H}_\dashv) \geq \eta_{\min}$, then the algorithm's auditing complexity is $\Omega(\ln(\frac{1-\delta}{\delta})\ln(1/\eta_{\min}))$.*

In the next section show that there are classes with a significant gap between active and auditing complexities even without an upper bound on the error.

## 4 Axis Aligned Rectangles

A natural extension of thresholds to higher dimension is the class of axis-aligned rectangles, in which the labels are determined by a $d$-dimensional hyperrectangle. This hypothesis class, first introduced in Blumer et al. (1989), has been studied extensively in different regimes (Kearns, 1998; Long and Tan, 1998), including active learning (Hanneke, 2007b). An axis-aligned-rectangle hypothesis is a disjunction of $2d$ thresholds. For simplicity of presentation, we consider here the slightly simpler class of disjunctions of $d$ thresholds over the positive orthant $\mathbb{R}_+^d$. It is easy to reduce learning of an axis-aligned rectangle in $\mathbb{R}^d$ to learning of a disjunction of thresholds in $\mathbb{R}^{2d}$ by mapping each point $\mathbf{x} \in \mathbb{R}^d$ to a point $\tilde{\mathbf{x}} \in \mathbb{R}^{2d}$ such that for $i \in [d]$, $\tilde{x}[i] = \max(x[i], 0)$ and $\tilde{x}[i + d] = \max(0, -x[i])$). Thus learning the class of disjunctions is equivalent, up to a factor of two in the dimensionality, to learning rectangles[1]. Because auditing costs are asymmetric, we consider two possibilities for label assignment. For a vector $\mathbf{a} = (a[1], \dots, a[d]) \in \mathbb{R}_+^d$, define the hypotheses $h_{\mathbf{a}}$ and $h_{\mathbf{a}}^-$ by

$$h_{\mathbf{a}}(x) = 2\mathbb{I}[\exists i \in [d], x[i] \geq a[i]] - 1, \quad \text{and} \quad h_{\mathbf{a}}^-(x) = -h_{\mathbf{a}}(x).$$

Define $\mathcal{H}_\square = \{h_{\mathbf{a}} \mid \mathbf{a} \in \mathbb{R}_+^d\}$ and $\mathcal{H}_\square^- = \{h_{\mathbf{a}}^- \mid \mathbf{a} \in \mathbb{R}_+^d\}$. In $\mathcal{H}_\square$ the positive points are outside the rectangle and in $\mathcal{H}_\square^-$ the negatives are outside. Both classes have VC dimension $d$. All of our results for these classes can be easily extended to the corresponding classes of general axis-aligned rectangles on $\mathbb{R}^d$, with at most a factor of two penalty on the auditing complexity.

## 4.1 The Realizable Case

We first consider the pool setting for the realizable case, and show a sharp contrast between the auditing complexity and the active label complexity for $\mathcal{H}_\square$ and $\mathcal{H}_\square^-$. Assume a pool of size $m$. While the active learning complexity for $\mathcal{H}_\square$ and $\mathcal{H}_\square^-$ can be as large as $m$, the auditing complexities for the two classes are quite different. For $\mathcal{H}_\square^-$, the auditing complexity can be as large as $m$, but for $\mathcal{H}_\square$ it is at most $d$. We start by showing the upper bound for auditing of $\mathcal{H}_\square$.

**Theorem 4.1** (Pool auditing upper bound for $\mathcal{H}_\square$). *The auditing complexity of any unlabeled pool $S_u$ of size $m$ with respect to $\mathcal{H}_\square$ is at most $d$.*

*Proof.* The method is a generalization of the approach to auditing for thresholds. Let $h^* \in \mathcal{H}_\square$ such that $\mathrm{err}(S, h^*) = 0$. For each $i \in [d]$, order the points $x$ in $S$ by the values of their $i$-th coordinates $x[i]$. Query the points sequentially from largest value to the smallest (breaking ties arbitrarily) and stop when the first negative label is returned, for some point $\mathbf{x}_i$. Set $a[i] \leftarrow x_i[i]$, and note that $h^*$ labels all points in $\{\mathbf{x} \mid x[i] > a[i]\}$ positive. Return the hypothesis $\hat{h} = h_{\mathbf{a}}$. This procedure clearly queries at most $d$ negative points and agrees with the labeling of $h^*$. $\qquad\qquad\square$

It is easy to see that a similar approach yields an auditing complexity of $2d$ for full axis-aligned rectangles. We now provide a lower bound for the auditing complexity of $\mathcal{H}_\square^-$ that immediately implies the same lower bound for active label complexity of $\mathcal{H}_\square^-$ and $\mathcal{H}_\square$.

**Theorem 4.2** (Pool auditing lower bound for $\mathcal{H}_\square^-$). *For any $m$ and any $d \geq 2$, there is a pool $S_u \subseteq \mathbb{R}_+^d$ of size $m$ such that its auditing complexity with respect to $\mathcal{H}_\square^-$ is $m$.*

*Proof.* The construction is a simple adaptation of a construction due to Dasgupta (2005), originally showing an active learning lower bound for the class of hyperplanes. Let the pool be composed of $m$ distinct points on the intersection of the unit circle and the positive orthant: $S_u = \{(\cos\theta_j, \sin\theta_j)\}$ for distinct $\theta_j \in [0, \pi/2]$. Any labeling which labels all the points in $S_u$ negative except any one point is realizable for $\mathcal{H}_\square^-$, and so is the all-negative labeling. Thus, any algorithm that distinguishes between these different labelings with probability 1 must query all the negative labels. $\qquad\square$

**Corollary 4.3** (Realizable active label complexity of $\mathcal{H}_\square$ and $\mathcal{H}_\square^-$). *For $\mathcal{H}_\square$ and $\mathcal{H}_\square^-$, there is a pool of size $m$ such that its active label complexity is $m$.*

## 4.2 The Agnostic Case

We now consider $\mathcal{H}_\square$ in the agnostic case, where $\eta > 0$. The best known algorithm for active learning of rectangles $(2, \delta)$-learns a very restricted class of distributions (continuous product distributions which are sufficiently balanced in all directions) with an active label complexity of $\tilde{O}(d^3 p(\ln(1/\eta)) p(\ln(1/\delta)))$, where $p(\cdot)$ is a polynomial (Hanneke, 2007b). However, for a general distribution, active label complexity cannot be significantly better than passive label complexity. This is formalized in the following theorem.

**Theorem 4.4** (Agnostic active label complexity of $\mathcal{H}_\square$). *Let $\alpha, \eta > 0, \delta \in (0, \frac{1}{2})$. Any learning algorithm that $(\alpha, \delta)$-learns all distributions such that $\mathrm{err}(D, \mathcal{H}) = \eta$ for $\eta > 0$ with respect to $\mathcal{H}_\square$ has an active label complexity of $\Omega(d/\eta)$.*

In contrast, the auditing complexity of $\mathcal{H}_\square$ can be much smaller, as we show for Alg. 3 below.

**Theorem 4.5** (Auditing complexity of $\mathcal{H}_\square$). *For $\eta_{\min}, \alpha, \delta \in (0, 1)$, there is an algorithm that $(\alpha, \delta)$-learns all distributions with $\eta \geq \eta_{\min}$ with respect to $\mathcal{H}_\square$ with an auditing complexity of $O(\frac{d^2 \ln(1/\alpha\delta)}{\alpha^2} \ln^2(1/\eta_{\min}))$.*

If $\eta_{\min}$ is polynomially close to the true $\eta$, we get an auditing complexity of $O(d^2 \ln^2(1/\eta))$, compared to the active label complexity of $\Omega(d/\eta)$, an exponential improvement in $\eta$. It is an open question whether the quadratic dependence on $d$ is necessary here.

Alg. 3 implements a 'low-confidence' version of the realizable algorithm. It sequentially queries points in each direction, until enough negative points have been observed to make sure the threshold in this direction has been overstepped. To bound the number of negative labels, the algorithm iteratively refines lower bounds on the locations of the best thresholds, and an upper bound on the *negative error*, defined as the probability that a point from $D$ with negative label is classified as

positive by a minimal-error classifier. The algorithm uses queries that mostly result in positive labels, and stops when the upper bound on the negative error cannot be refined. The idea of iteratively refining a set of possible hypotheses has been used in a long line of active learning works (Cohn et al., 1994; Balcan et al., 2006; Hanneke, 2007a; Dasgupta et al., 2008). Here we refine in a particular way that uses the structure of $\mathcal{H}_\square$, and allows bounding the number of negative examples we observe.

We use the following notation in Alg. 3. The negative error of a hypothesis is $\mathrm{err}_{\mathrm{neg}}(D, h) = \mathbb{P}_{(X,Y)\sim D}[h(X) = 1 \text{ and } Y = -1]$. It is easy to see that the same convergence guarantees that hold for $\mathrm{err}(\cdot, \cdot)$ using a sample size $m_\nu(\epsilon, \delta, d)$ hold also for the negative error $\mathrm{err}_{\mathrm{neg}}(\cdot, \cdot)$ (see Sabato et al., 2013). For a labeled set of points $S$, an $\epsilon \leq (0, 1)$ and a hypothesis class $\mathcal{H}$, denote $V_\nu(S, \epsilon, \mathcal{H}) = \{h \in \mathcal{H} \mid \mathrm{err}(S, h) \leq \mathrm{err}(S, \mathcal{H}) + (2\nu + \nu^2) \cdot \max(\mathrm{err}(S, \mathcal{H}), \epsilon)\}$. For a vector $\mathbf{b} \in \mathbb{R}_+^d$, define $\mathcal{H}_\square[\mathbf{b}] = \{h_{\mathbf{a}} \in \mathcal{H}_\square \mid \mathbf{a} \geq \mathbf{b}\}$.

---

**Algorithm 3:** Auditing for $\mathcal{H}_\square$

1: **Input:** $\eta_{\min} > 0$, $\alpha \in (0, 1]$, access to distribution $D$ over $\mathbb{R}_+^d \times \{-1, +1\}$.
2: $\nu \leftarrow \alpha/25$.
3: **for** $t = 0$ to $\lfloor \log_2(1/\eta_{\min}) \rfloor$ **do**
4:     $\eta_t \leftarrow 2^{-t}$.
5:     Draw a sample $S_t$ of size $m_\nu(\eta_t, \delta/\log_2(1/\eta_{\min}), 10d)$ with hidden labels.
6:     **for** $i = 1$ to $d$ **do**
7:         $j \leftarrow 0$
8:         **while** $j \leq \lceil (1 + \nu)\eta_t|S_t| \rceil + 1$ **do**
9:             If unqueried points exist, query the unqueried point with highest $i$'th coordinate;
10:             If query returned $-1$, $j \leftarrow j + 1$.
11:         **end while**
12:         $b_t[i] \leftarrow$ the $i$'th coordinate of the last queried point, or 0 if all points were queried.
13:     **end for**
14:     Set $S_{\mathbf{b}_t}$ to $S_t$, with unqueried labels set to $-1$.
15:     $V_t \leftarrow V_\nu(S_{\mathbf{b}_t}, \eta_t, \mathcal{H}_\square[\mathbf{b}_t])$.
16:     $\hat{\eta}_t \leftarrow \max_{h \in V_t} \mathrm{err}_{\mathrm{neg}}(S_{\mathbf{b}_t}, h)$.
17:     **if** $\hat{\eta}_t > \eta_t/4$ **then**
18:         Skip to step 21
19:     **end if**
20: **end for**
21: Return $\hat{h} \equiv \mathrm{argmin}_{h \in \mathcal{H}_\square[\mathbf{b}_t]} \mathrm{err}(S_{\mathbf{b}_t}, h)$.

---

Theorem 4.5 is proven in Sabato et al. (2013). . The proof idea is to show that at each round $t$, $V_t$ includes any $h^* \in \mathrm{argmin}_{h \in \mathcal{H}} \mathrm{err}(D, h)$, and $\hat{\eta}_t$ is an upper bound on $\mathrm{err}_{\mathrm{neg}}(D, h^*)$. Further, at any given point minimizing the error on $S_{\mathbf{b}_t}$ is equivalent to minimizing the error on the entire (unlabeled) sample. We conclude that the algorithm obtains a good approximation of the total error. Its auditing complexity is bounded since it queries a bounded number of negative points at each round.

## 5 Outcome-dependent Costs for a General Hypothesis Class

In this section we return to the realizable pool setting and consider finite hypothesis classes $\mathcal{H}$. We address general outcome-dependent costs and a general space of labels $\mathcal{Y}$, so that $\mathcal{H} \subseteq \mathcal{Y}^{\mathcal{X}}$. Let $S \subseteq \mathcal{X}$ be an unlabeled pool, and let $\mathrm{cost} : S \times \mathcal{H} \to \mathbb{R}_+$ denote the cost of a query: For $x \in S$ and $h \in \mathcal{H}$, $\mathrm{cost}(x, h)$ is the cost of querying the label of $x$ given that $h$ is the true (unknown) hypothesis. In the auditing setting, $\mathcal{Y} = \{-1, +1\}$ and $\mathrm{cost}(x, h) = \mathbb{I}[h(x) = -1]$. For active learning, $\mathrm{cost} \equiv 1$. Note that under this definition of cost function, the algorithm may not know the cost of the query until it reveals the true hypothesis.

Define $\mathrm{OPT}_{\mathrm{cost}}(S)$ to be the minimal cost of an algorithm that for any labeling of $S$ which is consistent with some $h \in \mathcal{H}$ produces a hypothesis $\hat{h}$ such that $\mathrm{err}(S, \hat{h}) = 0$. In the active learning setting, where $\mathrm{cost} \equiv 1$, it is NP-hard to obtain $\mathrm{OPT}_{\mathrm{cost}}(S)$ for general $\mathcal{H}$ and $S$. This can be

shown by a reduction to set-cover (Hyafil and Rivest, 1976). A simple adaptation of the reduction for the auditing complexity, which we defer to the full version of this work, shows that it is also NP-hard to obtain $\mathrm{OPT}_{\mathrm{cost}}(S)$ in the auditing setting.

For active learning, and for query costs that do not depend on the true hypothesis (that is $\mathrm{cost}(x, h) \equiv \mathrm{cost}(x)$), Golovin and Krause (2011) showed an efficient greedy strategy that achieves a cost of $O(\mathrm{OPT}_{\mathrm{cost}}(S) \cdot \ln(|\mathcal{H}|))$ for any $S$. This approach has also been shown to provide considerable performance gains in practical settings (Gonen et al., 2013). The greedy strategy consists of iteratively selecting a point whose label splits the set of possible hypotheses as evenly as possible, with a normalization proportional on the cost of each query.

We now show that for outcome-dependent costs, another greedy strategy provides similar approximation guarantees for $\mathrm{OPT}_{\mathrm{cost}}(S)$. The algorithm is defined as follows: Suppose that so far the algorithm requested labels for $x_1, \ldots, x_t$ and received the corresponding labels $y_1, \ldots, y_t$. Letting $S_t = \{(x_1, y_1), \ldots, (x_t, y_t)\}$, denote the current version space by $V(S_t) = \{h \in \mathcal{H}|_S \mid \forall (x, y) \in S_t, h(x) = y\}$. The next query selected by the algorithm is

$$ x \in \operatorname*{argmax}_{x \in S} \min_{h \in \mathcal{H}} \frac{|V(S_t) \setminus V(S_t \cup \{(x, h(x))\})|}{\mathrm{cost}(x, h)}. $$

That is, the algorithm selects the query that in the worst-case over the possible hypotheses, would remove the most hypotheses from the version spaces, when normalizing by the outcome-dependent cost of the query. The algorithm terminates when $|V(S_t)| = 1$, and returns the single hypothesis in the version space.

**Theorem 5.1.** *For any cost function* $\mathrm{cost}$*, hypothesis class $\mathcal{H}$, pool $S$, and true hypothesis $h \in \mathcal{H}$, the cost of the proposed algorithm is at most $(\ln(|\mathcal{H}|_S - 1) + 1) \cdot \mathrm{OPT}$.*

If $\mathrm{cost}$ is the auditing cost, the proposed algorithm corresponds to the following intuitive strategy: At every round, select a query such that, if its result is a negative label, then the number of hypotheses removed from the version space is the largest. This strategy is consistent with a simple principle based on a partial ordering of the points: For points $x, x'$ in the pool, define $x' \preceq x$ if $\{h \in \mathcal{H} \mid h(x') = -1\} \supseteq \{h \in \mathcal{H} \mid h(x) = -1\}$, so that if $x'$ has a negative label, so does $x$. In the auditing setting, it is always preferable to query $x$ before querying $x'$. Therefore, for any realizable auditing problem, there exists an optimal algorithm that adheres to this principle. It is thus encouraging that our greedy algorithm is also consistent with it.

An $O(\ln(|\mathcal{H}|_S))$ approximation factor for auditing is less appealing than the same factor for active learning. By information-theoretic arguments, active label complexity is at least $\log_2(|\mathcal{H}|_S)$ (and hence the approximation at most squares the cost), but this does not hold for auditing. Nonetheless, hardness of approximation results for set cover (Feige, 1998), in conjunction with the reduction to set cover of Hyafil and Rivest (1976) mentioned above, imply that such an approximation factor cannot be avoided for a general auditing algorithm.

# 6   Conclusion and Future Directions

As summarized in Section 2, we show that in the auditing setting, suitable algorithms can achieve improved costs in the settings of thresholds on the line and axis parallel rectangles. There are many open questions suggested by our work. First, it is known that for some hypothesis classes, active learning cannot improve over passive learning for certain distributions (Dasgupta, 2005), and the same is true for auditing. However, exponential speedups are possible for active learning on certain classes of distributions (Balcan et al., 2006; Dasgupta et al., 2008), in particular ones with a small disagreement coefficient (Hanneke, 2007a). It is an open question whether a similar property of the distribution can guarantee an improvement with auditing over active or passive learning. This might be especially relevant to important hypothesis classes such as decision trees or halfspaces. An interesting generalization of the auditing problem is a multiclass setting with a different cost for each label. Finally, one may attempt to optimize other performance measures for auditing, as described in the introduction. These measures are different from those studied in active learning, and may lead to new algorithmic insights.

## Footnotes

[1]This reduction suffices if the origin is known to be in the rectangle. Our algorithms and results can all be extended to the case where rectangles are not required to include the origin. To keep the algorithm and analysis as simple as possible, we state the result for this special case.

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
