[Reviews · NeurIPS 2013]

Submitted by Assigned_Reviewer_5

A relevant and well-written technical paper, which presents a variation of the classic active learning setting coined ‘auditing’. This is generally defined by non-uniform cost of labels and not knowing the cost of a label a priori to the label query. The aim of the paper is to compare the complexity of auditing versus (standard) active learning, i.e., how many labels are required. The authors accomplishes this by deriving bounds on the complexity for the new variant and compares to standard active learning where they show interesting results.

The authors focuses on two (tractable) cases:
1) They consider the active learning complexity as the total number of label queries.
2) They consider the auditing complexity as the number of label queries on points with negative label.

The resulting technical analysis is very comprehensive (and not verified in every detail by the reviewer) but seems mathematically sound. Only thing missing is an empirical evaluation or visualization - a simple synthetic example visualizing the principles and/or bounds would have eased the understanding. A real dataset would have provided practitioners with “evidence” that the principle works in the real agnostic settings (not only threshold and rectangles); however, the authors’ priorities are understandable.

Clarity:
• The paper is very well-written and clearly formulated – and with a few more explanations it may even be accessible to non-experts in complexity analysis.
• The reference seems adequate.

Significance:
• The thorough technical analysis and comparisons seems of sufficiently interesting to a part of the NIPS community; however, the more practical part of the community may find the paper uninteresting and effectively useless since no actual application and empirical evaluation is included.
• The work points towards the analysis of more interesting active learning paradigms where cost and task may be different than the standard setting of uniform cost and generalization performance. That is, it would for example be very interesting to make a complexity analysis of the “expected improvement” principle often used in (semi) Bayesian variation of active learning.

Originality:
• While the variation of active learning may seem incremental, and analysis largely based on known techniques, the work seems sufficiently original and the theoretical results themselves are novel and interesting.

Detailed comments:
• The definition of the threshold and rectangle problem should be define first time it is referred to in section 2.
• In the notation section a few more comments regarding the sample size definitions (line 0.79), e.g. epsilon, delta and c) may provide the novice reader with a better understanding of the theoretical analysis.
• For readability (for non-expert in complexity theory), it may be beneficial to explain what the Omega notation means. Again, this will make the paper more accessible to non-experts – at least the main results.
• Line 115/116: be -> but (also please check whole sentence).
• I would like a short summary of the results in section 6 – and how it is relevant to any application - just to remind the reader of the results before outlining the future (perhaps simple a reference to section 2 ?).
• Since you refer directly to material in the supplementary material, it should appear in the reference list and reference explicitly, and further published with the paper (e.g. as technical report, arXiv, or the like).
• A formal short conclusion should be included.
Summary: The paper introduces ‘auditing’ as an interesting and relevant active learning paradigm. The paradigm is carefully analyzed theoretically in simple settings and shows interesting complexity advantages over standard active learning.

Submitted by Assigned_Reviewer_6

This paper considers the complexity of active learning in a particular problem
setting, in which the cost of a label is highly non-uniform: positive labels
are free but negative labels are not. This setting is relevant in
applications such as fraud detection, where annoying someone about a
non-fraudulent transaction is costly.

The main results in the paper show that auditing complexity can be
significantly better than more standard active learning, and propose some
modifications of existing algorithms to capitalize on the special structure of
auditing problems. These generally involve ordering the queries so that
positive labels are queried first, and the number of negative labels can be
tightly bounded. Complexity results and associated specialized algorithms for
auditing are given for several classifiers, such as thresholds, rectangles,
more general hypothesis classes.

The paper is clearly written, and appears to offer several new complexity
results and algorithms that are relevant to the important problem of active
learning, for this particular setting. Various cases concerning the
achievable error of the hypothesis class are carefully described and analyzed.
It appears to be technically sounds, but I did not verify the proofs.

One interesting question that the authors do not address is whether their
results apply to a less severe non-uniform case, such as where one label is
considerably more costly but neither are free. This would seem to be a more
natural and widely-applicable setting, such as in medical diagnosis or general
cascading approaches where false positives are much cheaper than misses. I do
not think their methods carry over to this setting, but it would be good to
address this, and more generally say more about the relevance of the problem
setting. This makes me less positive about the significance of the work.
Summary: This paper considers the complexity of active learning in a particular problem
setting, in which positive labels are free but negative labels are not. This
problem setting is limited in applicability, but the paper clearly and
carefully analyzes a variety of hypothesis classes and their achievable error,
and lays out the relationship between standard active learning and this
auditing setting, as well as adapting algorithms to this particular setting.

Submitted by Assigned_Reviewer_8

Overview:
The authors address the problem of auditing, a restriction of the more general class-dependent cost sensitive active learning for binary classification where one pays for receiving labels on just the ‘negative’ examples. The paper presents a number of theoretical results for ‘auditing complexity’ (the number of negative labels required). Firstly, they note that, like active learning, in the general case auditing cannot necessarily improve over passive learning. Their main result is to show improved auditing complexity over active label complexity for threshold and axis aligned rectangular decision boundaries (both in the realisable and noisy, agnostic case). Finally they present a general algorithm to adapt a greedy active learner to auditing achieving constant speedup and the same complexity as the learner of Golovin, Krause (2011).

Quality:
In this work they present new complexity results, and the algorithms by which they may be achieved. The theoretical content of the paper is thorough and the algorithms presented yield insight into the differences between the auditing and active learning problem. The main results, and algorithms provided, apply only to highly restricted settings that are unlikely to be of much use in practice. They show improved complexity of auditing only for data that can be classified with axis aligned thresholds, or rectangles - with a further restriction that the algorithm only works if the negative examples are on the inside of the rectangle. In the noisy case (as discussed in the paper) the algorithm must be provided with the noise level in advance. Therefore although the theoretical results are insightful, these restrictions appear to strong to be of practical relevance.

One concern I have is with the agnostic case for the rectangular classification boundary - although there is an exponential reduction in complexity with respect to the noise level \eta, there is a polynomial increase the dimension, d. It is not clear to me in practice whether this trade-off will necessarily be beneficial, further discussion of typical values (or even an empirical evaluation) would make the case that auditing complexity is better than the standard active sampling complexity more convincing.

Although the paper is theoretical in nature, I find the complete lack of experimental evaluation of their proposed algorithms disappointing. Even if only on toy data, if the authors could show empirically the improved auditing performance of their auditing algorithms over both passive and active sampling it would make the paper more impactful to the more general reader.

Clarity:
The paper is clearly written and intuitions are provided to assist the reader in understanding the high-level workings of the algorithms and exploitable structure for auditing in the scenarios addressed in the paper.

Originality:
Although auditing has been indirectly addressed as it is a specific instance of cost-sensitive active learning, addressing the problem directly seems novel and useful. The application to fraud presented in the introduction is convincing, perhaps the authors could provide a could more motivating examples here?

The authors claim that in existing work in cost-sensitive active learning the costs need to be known in advance - however, for example, in the work of Kapoor et al (2007) the cost may be class dependent which is unknown at time of sampling, they take expectations of the cost function to account for this uncertainty. Surely their method could be directly applied to the auditing scenario too?

Significance:
The paper provides new theory on auditing, which is a relatively unstudied area. Although I feel that the algorithms provided have little practical significance the results presented in this paper yield insight into the auditing problem and could provoke further study into the practical side of active auditing.
Summary: This paper provides some new theoretical results for auditing; this appears to be a novel regime, and their results provide insight into the problem. However, the algorithms provided address only to restrictive settings and seem unlikely to be practically useful. The lack of empirical study also makes the study less convincing.
Author Feedback

Author rebuttal: Dear Reviewers,
We truly appreciate your thorough reviews and encouraging words and have taken into account all of your comments in the preparation of the final manuscript. Please see below our response to specific questions.


Reviewer 5

Thank you for the detailed comments, we have made the suggested changes to improve clarity.

A more general analysis of learning with imbalanced costs is certainly interesting (also discussed by Reviewer 6). We believe that our methods can serve as a starting point for addressing the more general setting.


Reviewer 6

We agree that the methods proposed in this work are not optimal for the general imbalanced cost case. We do believe that our results for the extreme case of auditing and our comparison to regular active learning can be helpful in defining the parameters for the general cost setting.


Reviewer 8

Regarding the polynomial increase in the dependence on the dimensions for axis parallel rectangles, this issue also comes up in regular active learning in the agnostic case, such as active learning for APRs and hyperplanes, thus it is possibly unavoidable. The implications are that auditing, like active learning, provides significant gains when the noise level is low. In particular, our results for auditing APRs are significant when the noise level is o(1/d).

The 2007 IJCAI paper by Kapoor et al. addresses varying error and query costs more generally, and they indeed mention that if the query cost depends on the class of the point, then the expected cost (based on an estimate of the probability of each label) can be used instead (this is mostly suitable for probabilistic classifiers). Because the focus of their work is not on class-dependent costs, they neither provide an analysis of that scenario nor study differences versus standard active learning. This makes our work complementary to theirs, and we have added a relevant note to our paper. Thanks!